

# Fascin actin-bundling protein 1 regulates non-small cell lung cancer progression by influencing the transcription and splicing of tumorigenesis-related genes

Qingchao Sun, Ruixue Liu, Haiping Zhang, Liang Zong, Xiaoliang Jing, Long Ma, Jie Li and Liwei Zhang

Department of Thoracic Surgery, The First Affiliated Hospital of Xinjiang Medical University, Urumqi, Xinshi District, China

Corresponding author
Liwei Zhang, zhangliweixj@163.com

## ABSTRACT

**Background**. High mortality rates are prevalent among patients with non-small-cell lung cancer (NSCLC), and effective therapeutic targets are key prognostic factors. Fascin actin-bundling protein 1 (FSCN1) promotes NSCLC; however, its role as an RNA-binding protein in NSCLC remains unexplored. Therefore, we aimed to explore FSCN1 expression and function in A549 cells.

**Method**. We screened for alternative-splicing events and differentially expressed genes (DEGs) after *FSCN1* silence *via* RNA-sequencing (RNA-seq). FSCN1 immunoprecipitation followed by RNA-seq were used to identify target genes whose mRNA expression and pre-mRNA alternative-splicing levels might be influenced by FSCN1.

**Results**. Silencing *FSCN1* in A549 cells affected malignant phenotypes; it inhibited proliferation, migration, and invasion, and promoted apoptosis. RNA-seq analysis revealed 2,851 DEGs and 3,057 alternatively spliced genes. Gene ontology-based functional enrichment analysis showed that downregulated DEGs and alternatively splicing genes were enriched for the cell-cycle. FSCN1 promoted the alternative splicing of cell-cycle-related mRNAs involved in tumorigenesis (*i.e.*, *BCCIP*, *DLGAP5*, *PRC1*, *RECQL5*, *WTAP*, and *SGO1*). Combined analysis of FSCN1 RNA-binding targets and RNA-seq data suggested that FSCN1 might affect *ACTG1*, *KRT7*, and *PDE3A* expression by modulating the pre-mRNA alternative-splicing levels of *NME4*, *NCOR2*, and *EEF1D*, that were bound to long non-coding RNA transcripts (*RNASNHG20*, *NEAT1*, *NSD2*, and *FTH1*), which were highly abundant. Overall, extensive transcriptome analysis of gene alternative splicing and expression levels was performed in cells transfected with FSCN1 short-interfering RNA. Our data provide global insights into the regulatory mechanisms associated with the roles of FSCN1 and its target genes in lung cancer.

## INTRODUCTION

Lung cancer is the leading cause of cancer-related deaths worldwide (*Cao & Chen, 2019*), and its incidence is continually increasing. Only 25% of patients with non-small-cell lung cancer (NSCLC) undergo early-stage radical surgery, but only 50–70% of them have a

5-year survival rate (*Oliver, 2022*). Half of patients with NSCLC experience relapse due to resistance to chemotherapy (*Nooreldeen & Bach, 2021*). The early diagnosis of patients with NSCLC, as well as the assessment of their chemosensitivity or radiosensitivity, requires urgent improvement. Exploring the molecular mechanisms underlying NSCLC development could help identify novel therapeutic targets and support NSCLC molecular diagnosis and the development of targeted therapeutics.

There are approximately 1,914 human RNA-binding proteins (RBPs) that function as protein-coding genes (*Neelamraju, Hashemikhabir & Janga, 2015*). RBPs regulate almost every step of RNA expression and help maintain cellular homeostasis (*Matia-González, Laing & Gerber, 2015*). Numerous studies have confirmed that RBPs regulate the expression and function of oncogenes and tumor suppressor genes by mediating RNA modification in different types of tumor progression (*Lujan, Ochoa & Hartley, 2018*). The overexpression of fascin actin-binding protein 1 (FSCN1) has been associated with tumor malignant phenotypes (*Liu et al., 2021*). A recent study on esophageal cancer reported that FSCN1 may function as an RBP; FSCN1 binding to the PTK6 mRNA precursor was found to inhibit PTK6 transcription (*Cai et al., 2022*). The molecular mechanism(s) underlying the role of FSCN1 as an RBP in NSCLC remains unknown.

Therefore, we aimed to explore FSCN1 expression and function in A549 cells. Specifically, we silenced *FSCN1* in A549 cells, which inhibited proliferation, invasion, and migration, and promoted apoptosis. High-throughput RNA-sequencing (RNA-seq) was performed to identify differentially expressed genes (DEGs) and alternatively splicing genes (ASGs) after FSCN1 knockdown, and the associated pathways were explored. iRIP-Seq data showed that FSCN1 performs specific functions together with significant long noncoding RNAs (lncRNAs). We also performed association analysis between FSCN1 binding target RNA and RNA-Seq to identify genes and alternative splicing levels that FSCN1 binds and regulates. In this study, FSCN1 was regarded as an RBP and a breakthrough point to explore the possible role of FSCN1 in gene transcription and alternative splicing (AS), thereby providing insight into the future diagnosis and treatment of NSCLC.

## MATERIALS & METHODS

### FSCN1 expression in tumor samples
We used Gene Expression Profiling Interactive Analysis (GEPIA2) and Kaplan–Meier (K–M)-plotter web-based tools to analyze *FSCN1* expression, disease-free progression, and overall survival based on the public database, The Cancer Genome Atlas (TCGA). Patient information used in this study was pathologically staged according to the TNM staging of the eighth edition of the American Joint Committee on Cancer (AJCC). The correlation between the tumor stage and FSCN1 expression was evaluated.

### Cell culture
A549-1, A549-2, HTR-8, and Huh7 cells were purchased from ProCell Life Science & Technology Co., Ltd. (Jiangsu, China) and cultured in Ham's F-12K (ProCell Life Science & Technology) medium containing 1% streptomycin/penicillin (HyClone, Logan, UT, USA) and 10% fetal bovine serum (FBS) (Gibco, Shanghai, China). The cells were incubated

at 37 °C in an incubator containing 5% $CO_2$ with 85–90% humidity. A Mycoplasma Detection Kit (MD001; Shanghai Yise Medical Technology Co., Ltd., Shanghai, China) was used to assess the presence of mycoplasma contamination in A549 cells. The background expression of *FSCN1* in A549 cells was evaluated using western blotting.

### A549 cell plasmid transfections

A549 cells were transfected at the logarithmic growth phase and the degree of confluence ranged from 60–70% in 12-well plates. The FSCN1-specific siRNA were siFSCN1-1926 (5′-GAGCCUUAUUUCUCUGGAATT- 3′), siFSCN1-1122 (5′-GAAUGCCAGCUGCUA CUUUTT- 3′), and siFSCN1-556 (5′-GUCAACAUCUACAGCGUCATT- 3′); the scramble siRNA (NC) was siNegative (5′-UUCUCCGAACGUGUCACGUTT- 3′). We performed transfection using Lipofectamine™ RNAiMAX Transfection Reagent (13778030; Invitrogen, Waltham, MA, USA) according to the manufacturer's protocol. All transfected cells were collected at 48 h for subsequent experiments.

### RT-qPCR validation of FSCN1 silencing in A549 cells

We extracted total RNA from A549 cells according to the TRIzol manufacturer's protocol and tested RNA quality and quantity using the ultra-micro spectrophotometer (NanoPhotometer N50; Implen, München, Germany). RNA was reverse transcribed using HiScript III RT SuperMix for qPCR (+gDNA wiper; R323-01; Vazyme, Xuanwu Qu, China). cDNA was used as the template. RT-qPCR analysis was conducted on the Bio-Rad S1000 with Hieff™ qPCR SYBR® Green Master Mix (Low Rox Plus; Yeasen, Shanghai, China) with heating to 94 °C for 15 s and 40 cycles of 94 °C for 5 s and 58 °C for 34 s. The mRNA level of each target gene was normalized to the glyceraldehyde-3-phosphate dehydrogenase (GAPDH) mRNA level using the $2 - \Delta\Delta CT$ method (*Livak & Schmittgen, 2001*). Expression differences were analyzed using a paired student's *t*-test with Prism (version 8.0; GraphPad, La Jolla, CA, USA). Details of the primers and samples are listed in Tables S1 and S2.

### Western blot assay to assess FSCN1 expression

A549 cells were lysed in ice-cold RIPA buffer (Proteintech, China) supplemented with a protease inhibitor cocktail (Roche, Basel, Switzerland) and incubated for 30 min. The protein concentration of the extracted A549 cells was determined according to the instructions of the BCA protein quantification kit (Solarbio, Beijing, China). Subsequently, the samples were loaded at different concentrations (20 μg per well). The samples were boiled in water with protein loading buffer (Solarbio) for 10 min. All target protein samples were resolved by constant pressure electrophoresis (60 V for 30 min, 120 V for 60 min) in 10% sodium dodecyl sulfate-polyacrylamide gel and transferred to 0.45-mm polyvinylidene fluoride (PVDF) membranes (Millipore, Burlington, MA, USA) under a constant current (0.35–0.45 A). The PVDF membrane was washed with TBST for 10 min, and then sealed with 5% BSA for 2 h on a shaker and incubated for 2 h at 26 °C with a mouse anti-FSCN1 antibody (1:2000; ABclonal, China) or a rabbit anti-GAPDH antibody (1:2000; Proteintech, Rosemont, IL, USA). PVDF membranes were then washed with TBST thrice (5 min each time) and incubated with a horseradish peroxidase-conjugated

secondary antibody (anti-rabbit, 1:7000, Proteintech; anti-mouse, 1:7000, ABclonal) for 45 min at 26 °C. Finally, the PVDF membrane was placed in enhanced ECL reagent (P0018FM; Beyotime, China) and the protein signals were detected using chemiluminescence (5200; Tanon, Shanghai, China).

## CCK-8 assay

We used CCK-8 (Yeasen) to conduct cell viability assays. In summary, control and experimental cells were treated accordingly, and vials without cells were used as blank controls. A549 cells were resuspended in medium and seeded at a density of 5,000 cells/well in 96-well plates. Following culture for 0, 24, 48, or 72 h at 37 °C and 5% $CO_2$, 10 µl CCK-8 solution was added to the medium at the corresponding time points, and the cells were incubated for 2 h. Optical densities (ODs) were measured at 450 nm using a microplate reader (Biotek, Winooski, VT, USA).

## Apoptosis assay

Apoptosis was detected *via* flow cytometry (FCM). A549 cells were transfected with a siRNA for 48 h. We used an Annexin V-APC/7-ADD Apoptosis Detection Kit (Yeasen). Then, 5 µl Annexin V-APC solution was added to the treated and control cells, which were incubated at 26 °C in the dark for 5 min. The treated and control cells were mixed with 5 µl 7-AAD reagent and incubated again for 5 min. More than 10,000 cells were analyzed through FCM (FACSCanto; BD Biosciences).

## Cell migration assay

Cells ($4 \times 10^4$) were added to 200 µl 0% FBS medium un Transwell chambers with 8-µm filters (Corning, Corning, NY, USA). The chambers were inserted into 600-µl medium in the lower chamber containing 10% FBS. Three replicates were set up for each group and incubated for 24 h at 37 °C and 5% $CO_2$. Migrated cells in the lower chamber were fixed for 20 min in 4% paraformaldehyde (Beyotime, Jiangsu, China), stained with 0.1% crystal violet (Beyotime), washed with PBS three times, and air dried in an ultra-clean chamber. Finally, cells were counted under an inverted microscope (MF52-N; Mshot, Guangzhou, China) at 200 × magnification.

## Cell invasion assay

The Transwell chambers with 8 µm filters were precoated with 100 µl of Matrigel (diluted 1:8 in 0% FBS medium; BD Biosciences, Franklin Lakes, NJ, USA) and incubated for 1 h at 37 °C and 5% $CO_2$. Subsequently, $1.2 \times 10^5$ cells were added to 200 µl 0% FBS medium to Transwell chambers. The chambers were inserted into 600 µl medium in the lower chamber containing 10% FBS. The chambers were incubated for 24 h. Cells remaining on the upper membrane surface were removed, and the invasive cells were counted under an inverted microscope (MF52-N; Mshot) at 200 × magnification.

## RNA-seq

Total RNA was extracted from A549 cells using the TRIzol Reagent (Invitrogen) as described by *Chomczynski & Sacchi (2006)*. DNase I was used to digest DNA. RNA integrity

was verified using 1.5% agarose gel electrophoresis. The RNA samples were quantified using a Qubit 3.0 instrument with the QubitTM RNA Broad Range Assay Kit (Life Technologies).

RNA-seq libraries we prepared using total RNA (2 μg) and the KCTM Stranded mRNA Library Prep Kit for Illumina (Wuhan Seqhealth, Wuhan, China). PCR products (200–500 base pairs [bp]) were applied to the NovaSeq 6000 sequencer (Illumina) using the PE150 model. PE150 represents the sequencing read length of 150 bp at both ends. Our sequencing depth is >10 G (sequencing volume 10–13 G), which can cover the human transcriptome in a relatively saturated way.

### Filtering and aligning raw RNA-seq data

Raw reads containing more than two "Ns" were discarded. Adaptors and low-quality bases were trimmed from the raw sequencing reads using the FASTX Toolkit (version 0.0.13). Reads below 16 nt in length were also excluded. Subsequently, clean reads were aligned to the GRCh38 genome using HISAT2, allowing for four mismatches (*Kim, Langmead & Salzberg, 2015*). Uniquely mapped reads were used to count gene reads and fragments per kilobase (kb) of transcript per million fragments mapped (FPKM) values (*Trapnell et al., 2010*). The replication and experimental power were calculated using the RNASeqPower package in R. The final power value of 0.97 was used to detect a two-fold expression.

### DEG analysis

We used the package DESeq2 to model the raw reads and the scale factor to account for library depth differences. DESeq2 can analyze the differential expression between two or more samples (cutoff criteria: $P < 0.05$; fold-change [FC] of >2 or <0.5) (*Love, Huber & Anders, 2014*).

### Alternative-splicing analysis

ABLas (*Xie et al., 2011*; *Jin et al., 2017*) was used to detect the types of alternative splicing events (ASEs) based on splice-junction reads. Unpaired two-tailed student's $t$-test was performed to assess RBP-regulated ASEs. Events that were significant at the $P$-value cutoff corresponding to a false-discovery rate of 5% were considered to be RBP-regulated ASEs.

### Functional enrichment analysis

Gene Ontology (GO) terms were identified using KOBAS 2.0 (*Xie et al., 2011*). The hypergeometric test and Benjamini–Hochberg FDR control procedures were used to define the enrichment for each term.

### Co-immunoprecipitation (co-IP) and library preparation

A549 cells were irradiated once at 400 mJ/cm$^2$ andlysed in ice-cold wash buffer containing 200 U/ml RNase inhibitor (Takara) and a protease-inhibitor cocktail (Roche), then incubated on ice for 30 min. Lysates were centrifuged at 10,000 × $g$ for 10 min at 4 °C. RQ I (Promega, 1 U/μl) was added to each supernatant (final concentration: 1 U/μl); the samples were incubated at 37 °C for 30 min, and then stop solution (EDTA, Thermo Scientific) was added. The mixtures were subjected to the usual protocol to remove cellular debris. RNA digestion was performed using MNase (Thermo Fisher Scientific, Waltham, MA, USA). Co-immunoprecipitation assays were performed using the Thermo

Scientific Pierce Co-IP kit (Thermo Fisher Scientific), according to the manufacturer's instructions. Western blot was used to detect FSCN1-antibody (Abcam) or control-IgG antibody (ABclonal, Woburn, MA, USA) of the samples. RNA was purified using phenol: chloroform: isopentyl alcohol (25:24:1, pH < 5; Solarbio). cDNA libraries were prepared using the KAPA RNA HyperPrep Kit (KAPA Biosystems) and used for high-throughput sequencing (150-nucleotide (nt) paired-end sequencing) with the Illumina NovaSeq system.

## Data analysis

We aligned the reads to the genome using HISAT2 (*Kim, Langmead & Salzberg, 2015*) to generate a unique genome comparison, excluding PCR duplicates. ABLIRC (*Xia et al., 2017*) and Piranha (*Uren et al., 2012*) were used for peak calling. ABLIRC enabled identification of binding regions in the human genome (GRCh38), version 38 (Ensembl 104) (*Xia et al., 2017*). For peak calling, the whole genome was first scanned with a 5 bp window in 5 bp steps. Peaks were identified when the depth of the first window was 2.5 times larger for eight consecutive windows or the medium depth was >50. Peaks ended when the depths of eight consecutive windows were < 4% of the maximum depth. The reads for each gene were randomly distributed to each gene 500 times, and the peak-depth frequency was determined for each peak, enabling screening for significant peaks ($P < 0.05$) or peaks with a maximum depth of $\geq 10$. Using input samples as controls, we conducted abundance-difference analysis for the peak locations, and for peaks with an abundance FC of >4 following IP (an adjustable parameter) the input abundance was screened as the final combination peak. The target genes identified by IP were determined based on the peaks, and the binding motifs of the immunoprecipitated proteins were called using HOMER software (*Heinz et al., 2010*).

## LncRNA predictions

We used the CPC2 (*Kong et al., 2007*), LGC (*Wang et al., 2019a*), CNCI (*Sun et al., 2013*), and CPAT (*Wang et al., 2013a*) software packages to identify and count credible lncRNAs. Subsequently, we removed transcripts that overlapped with known coding genes, and that were <200-bp long, potentially coding, and <1,000 bp away from the nearest gene. We obtained prediction results for new lncRNAs and used the intersecting results from the four software packages for subsequent analysis and processing.

## Cis-regulatory targets of lncRNAs

We set the co-location threshold as 100-kb upstream or downstream of a lncRNA in each trans-regulatory relationship pair (*Yang et al., 2016*) and then calculated Pearson correlation coefficients between lncRNAs and mRNAs for co-location and co-expression analyses to screen lncRNA-target relationship pairs with a correlation coefficient of $| > 0.6|$ and $P \leq 0.01$. We used the intersection of the co-location and co-expression datasets to identify cis targets of different lncRNAs.

## RAW sequencing data

Sequencing data have been uploaded to the NCBI (National Center for Biotechnology Information) sequence read archive GSE234860.

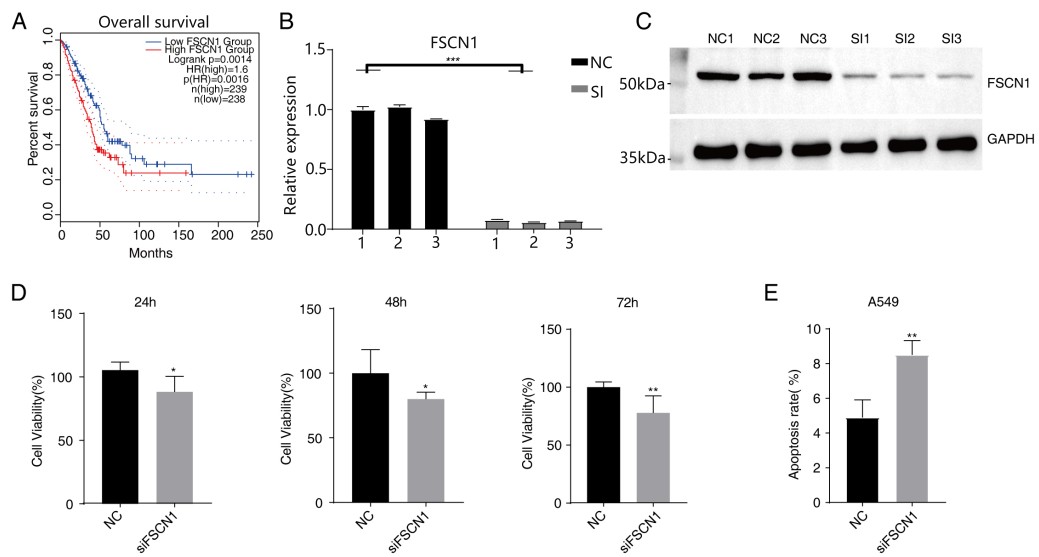

**Figure 1** **Knockdown of FSCN1 inhibits cellular proliferation and promotes apoptosis in A549 cells.** (A) The survival curve of FSCN1 in lung adenocarcinoma in TCGA database. (B) The histogram shows the RT-qPCR results of NC and SI samples. Error bars represent mean ± SEM. (C) Western blot experiment results showed that the FSCN1 knockdown was successful. (D) Proliferation results of A549 after FSCN1 knockdown. (E) Apoptosis results of A549 after FSCN1 knockdown. *P-value < 0.05, **P-value < 0.01 and ***P-value < 0.001.

## RESULTS

### Expression of FSCN1 in various tumors based on TCGA database

We used GEPIA2 and K–M methods to analyze gene expression levels and survival times based on TCGA data. Among the 31 tumor types studied, FSCN1 was highly expressed in 16 tumor types ($P < 0.05$), whereas FSCN1 expression was significantly lower in acute myeloid leukemia tissues ($P < 0.05$). FSCN1 was highly expressed in lung squamous cell carcinoma (LUSC, $P < 0.05$) but not in lung adenocarcinoma (LUAD). The changes in FSCN1 expression differed significantly across the four disease stages of LUAD ($P = 0.0166$) but not in LUSC. K–M analysis revealed that high FSCN1 expression predicted poor prognosis in patients with LUAD ($P = 0.0016$) but not in patients with LUSC ($P = 0.48$; Fig. S1 and Fig. 1A). However, high FSCN1 expression was not associated with disease-free progression in either LUSC or LUAD ($P > 0.05$; Fig. S2). Systematic reviews and meta-analyses have revealed that FSCN1 overexpression promotes metastasis and mortality in various tumors, suggesting that FSCN1 can be a therapeutic target for cancer (*Tan et al., 2013*). Further studies on FSCN1 are necessary as its characteristics and mechanism in NSCLC are not completely understood.

### FSCN1 inhibited cell proliferation, invasion, and migration and promoted apoptosis *in vitro*

Western blot analysis results showed that FSCN1 was highly expressed in A549 cells (Fig. S3). We transfected A549 cells with a plasmid that expressed FSCN1-siRNA (siFSCN1-556,

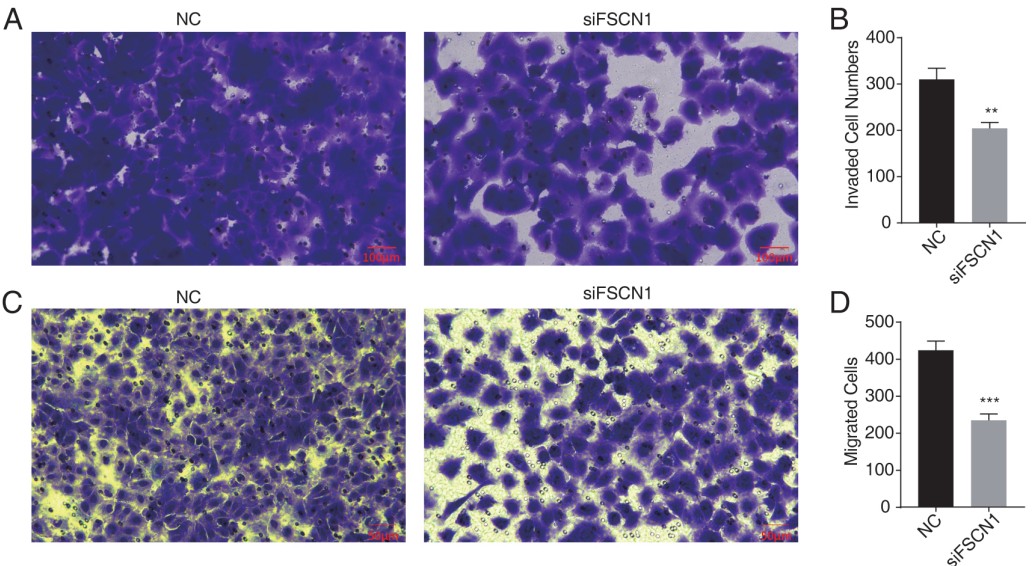

**Figure 2** **Knockdown of fascin actin-bundling protein 1 (FSCN1) inhibits cellular migration and invasion in A549 cells.** (A–B) The invasion results of A549 after FSCN1 knockdown. (C–D) The migration results of A549 after FSCN1 knockdown. **P-value < 0.01 and ***P-value < 0.001. NC, scramble siRNA (siNegative).

siFSCN1-1122, siFSCN1-1926 and scramble siRNA [NC]). RT-qPCR and western blotting showed that the expression of FSCN1 was significantly decreased ($P < 0.05$) in the FSCN1-siRNA transfection group than that in the control transfection group (Figs. 1B and 1C and Fig. S4). siFSCN1-1926 was used for proliferation, apoptosis, and cell invasion experiments; siFSCN1-1122 was used for metastasis experiments. CCK8 assays showed that silencing *FSCN1* significantly inhibited cell proliferation ($P < 0.05$, Fig. 1D), whereas flow cytometry revealed that silencing FSCN1 significantly increased apoptosis (Fig. 1E). Transwell assays showed that silencing FSCN1 in A549 cells resulted in significantly lower cell invasion and metastasis than in the control treatment (Fig. 2).Therefore, we successfully constructed an FSCN1 knockdown model in A549 cells, and demonstrated through *in vitro* experiments that FSCN1 knockdown suppressed the proliferation, invasion, and migration abilities of A549 cells and promoted their apoptosis. Thus, FSCN1 can be a potential target for NSCLC treatment.

## FSCN1 knockdown in A549 cells resulted in transcriptional differences

In this study, six cDNA libraries (three biological replicates for each library) were constructed based on siFSCN1 (siFSCN1-1926) and control transfectors, and the role of FSCN1 in the transcriptional regulation of A549 cells was explored using RNA-seq analysis. The FSCN1 FPKM value was significantly lower for the siFSCN1 transfectants than for the control groups ($P < 0.001$, Fig. 3A). Principal component analysis of all gene expression values revealed significant differences between the siFSCN1 and control groups (Fig. 3B). FCs of $\geq 2$ or $\leq 0.5$ (corrected $P < 0.05$) were used as criteria, FSCN1 silencing resulted in the significant differential expression of 2851 genes at the transcriptional level,

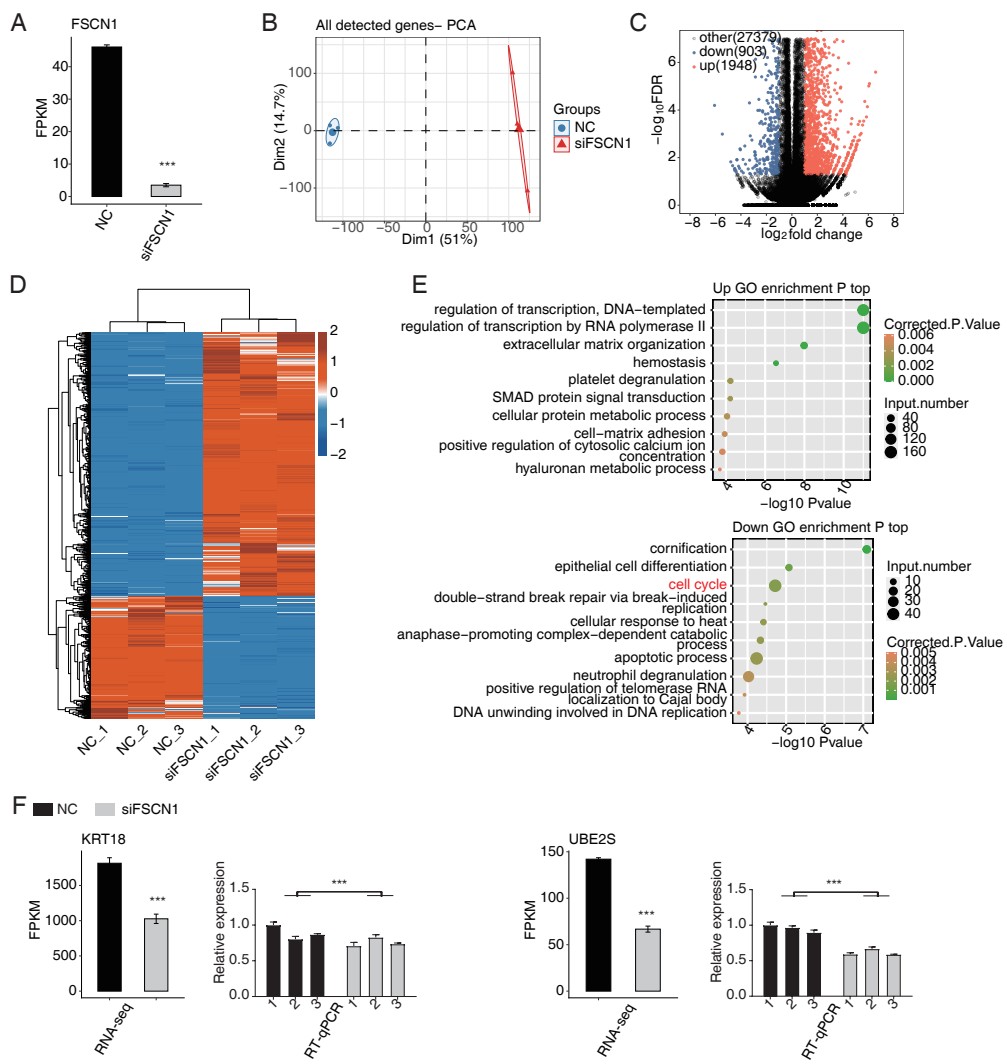

**Figure 3** **FSCN1 regulates gene expression of the cell cycle in A549 cells.** (A) Bar plot showing the expression pattern and statistical difference of FSCN1. Error bars represent mean ± SEM. ***$P$-value < 0.001. (B) PCA based on FPKM value of all detected *FSCN1* gene knockdowns. The ellipse for each group represent the confidence ellipse. (C) Volcano plot showing all differentially expressed genes (DEGs) between siFSCN1 and NC samples. (D) Hierarchical clustering heat map showing expression levels of all DEGs. (E) Scatter plot exhibiting the most enriched GO biological process results of the upregulated (top panel) and downregulated (bottom panel) DEGs. (F) Bar plot showing the expression pattern and statistical difference of DEGs for specific key genes. Reverse transcription qPCR validation of DEGs regulated by FSCN1 in cancer cells; black bars denote the control group and grey bars denote the FSCN1-silenced group. Error bars represent mean ± SEM. ***$P$-value < 0.001.

which included significantly more upregulated genes than downregulated genes after *FSCN1* knockdown (Fig. 3C).

Heat map analysis of the DEG expression patterns revealed similar gene expression patterns of the samples and close clustering relationships in the control and FSCN1-knockdown groups (Fig. 3D). GO enrichment analysis revealed that upregulated DEGs

in the *FSCN1*-knockdown group were mainly enriched for biological process terms such as DNA template transcription regulation, RNA polymerase II transcription regulation, extracellular matrix organization, hemostasis, platelet degranulation, SMAD protein signal transduction, cellular protein metabolism, cell matrix adhesion, positive regulation of plasma calcium concentration, and hyaluronic acid metabolism process, among others. The downregulated DEGs were mainly enriched for keratinization, epithelial cell differentiation, cell cycle, double-strand break repair induced by break, cell response to heat, late promotion of complex-dependent catabolic process, apoptosis process, neutrophil degranulation, positive regulation of telomerase RNA localization in CajaI, DNA in DNA replication (Fig. 3E). These results suggested that FSCN1 knockdown in A549 cells resulted in transcriptional differences in numerous genes, and that the functions of DEGs were closely related to tumor progression.

## FSCN1 regulates the expression levels of a large number of genes in A549 cells

According to the significances of the expression difference ratios between the experimental and control groups, we selected several upregulated DEGs (*LBH*, *IGFBP3*, *CDH1*, *TFPI*, and *IGFBP7*) and down regulated DEGs (*KRT18*, *UBE2S*, *UBE2C*, *CDC25B*, *MCM7*, and *CDK4*) for further analysis ($P < 0.05$; [FC] of >2 or $\leq$0.5) (Fig. 3F and Fig. S5). The upregulated DEGs have been associated with lung cancer growth, invasion, metastasis, prognosis, and drug resistance (*Amirkhosravi et al., 2002*; *Chen et al., 2009*; *Wang et al., 2013b*; *Liang et al., 2015*; *Deng et al., 2018*). Consistent with these findings, the DEGs were significantly downregulated after *FSCN1* knockdown correlated positively with FSCN1 expression and correlated with the growth, metastasis, and prognosis of NSCLC (*Boldrini et al., 2007*; *Liu et al., 2012*; *Zhang et al., 2014*; *Liu & Xu, 2018*; *Guo et al., 2018*). We explored the DEG expression levels and survival rates of patients with LUAD based on the GEPIA2 database. Compared with normal tissues, IGFBP7, LBH and TFPI are low expressed in tumor tissues ($P < 0.05$), while KRT18, UBE2C and UBE2S were highly expressed in tumor tissues ($P < 0.05$). K–M analysis showed that patients with elevated ACTG1, MCM7, KRT7, UBE2C, and UBE2S expression in lung tumors had shorter survival times ($P < 0.05$, Fig. S6). In addition, through PCR verification, we found that the expression levels of *UBE2S*, *KRT18*, and *ACTG1* in the experimental group were decreased compared with those in the control group, whereas the expression of *PDE3A* was increased; two-way ANOVA showed significant differences, as anticipated (Fig. 3F and Fig. S7). Therefore, we conclude that FSCN1 regulates the expression of genes related to the tumor malignant phenotype in A549 cells, indicating a potential regulatory mechanism.

## FSCN1 regulated the AS of cell cycle-related genes in A549 cells

Our main objective was to analyze ASEs in target genes regulated by FSCN1 using A549 cells. We evaluated splice junctions using RNA-seq data and the ABL as pipeline, which revealed ASEs in 3057 DEGs. A positive T-value indicated that the proportion of the splice type was higher in the siFSCN1 group than in the control group, and a negative T-value indicated the opposite (Fig. 4A). We identified 278 exon cassettes with increased numbers

of ASEs: 74 MXEs, 162 ESs, 50 A5SS and Es, 347 A5SSs, 47 A3SS and Es, 275 A3SSs, 77 5pMXEs, and 113 3pMXEs. We also identified 175 exon cassettes with decreased ASEs: 89 MXEs, 304 ESs, 69 A5SS and Es, 272 A5SSs, 58 A3SS and Es, 399 A3SSs, 142 5pMXEs, and 126 3pMXEs. Principal component analysis of the RASEs revealed separation between the siFSCN1 and control samples (Fig. 4B). Hierarchical clustering of the RASEs suggested similar expression patterns among samples within the group (Fig. 4C). GO functional enrichment analysis showed that RASEs were mainly enriched for RNA splicing, mRNA processing, mRNA splicing through spliceosomes, translation, cytoplasmic translation, DNA repair, REDOX process, tRNA processing, cell cycle, and cell division (Fig. 4D). In particular, the cell-cycle-related genes *PRC1*, *SGO1*, and *WTAP* were alternatively spliced. ES events in *PRC1*, 3pMXEs in *RECQL5*, and A5SSs in *SGO1* were significantly more frequent in the siFSCN1 group than in the control group (Fig. 4E and Fig. S8). In contrast, A3SSs in *WTAP* and ESs in *DLGAP5* were significantly less frequent in the *FSCN1*-knockdown group than in the control group (Fig. 4F and Fig. S8). The PCR results indicated that the rate of ASEs of *SGO1* in the experimental group was increased, whereas that of *WTAP* was decreased compared with the control group; two-way ANOVA showed significant differences, as anticipated (Figs. 4E and 4F). FSCN1 can mediate cancer cell malignant phenotype (*Chen et al., 2019*; *Liao et al., 2022*; *Wang et al., 2022*). Based on this, we speculated that FSCN1 may promote the development of NSCLC by regulating the variable splicing of cycle-related genes. In addition, we performed an integrated analysis of regulated alternatively splicing genes (RASGs) and DEGs and identified 182 genes with significant differences in expression and AS levels. Therefore, we deduce that FSCN1 acts as an RBP to regulate the AS levels of cell cycle-related genes, which may play an important role in NSCLC proliferation.

### FSCN1 bound LncRNAs and regulation of their expression in A549 cells

In A549 cells, an antibody against FSCN1 was used for co-IP experiments, and more distinct bands were found with the IP group than with the input and IgG groups (Fig. 5A). IP followed by RNA-seq (iRIP-seq) was used to cluster samples based on the RNA-seq data. The results indicated highly similar samples, large correlation coefficients, and close clustering (Fig. 5B). The input sample was used as the background to perform peak calling for the RMDUP reads to obtain the specific binding peaks for the experimental samples, and the distributions of the binding peaks in each region of the reference genome were determined (Fig. 5C). The IP group showed more enrichment for intergenic regions than the input group, and motif analysis revealed that FSCN1 bound to GC-rich regions (Fig. 5D). We found that FSCN1 could strongly bind LncRNA-SNHG20, LncRNA-NEAT1, NSD2, and FTH1 (Figs. 5E and 5F and Fig. S9). *Li et al. (2019)* reported that SNHG20 is highly expressed in glioma and promotes cell proliferation by silencing P21. LncRNA-NEAT1 promotes metastasis in soft-tissue sarcomas, and mass spectrometry revealed that most of the interacting proteins were involved in RNA splicing regulation (*Huang et al., 2020*). The iRIP-seq data of FSCN1 indicated that it may directly bind to some LncRNAs to regulate splicing, which has scarcely been reported.

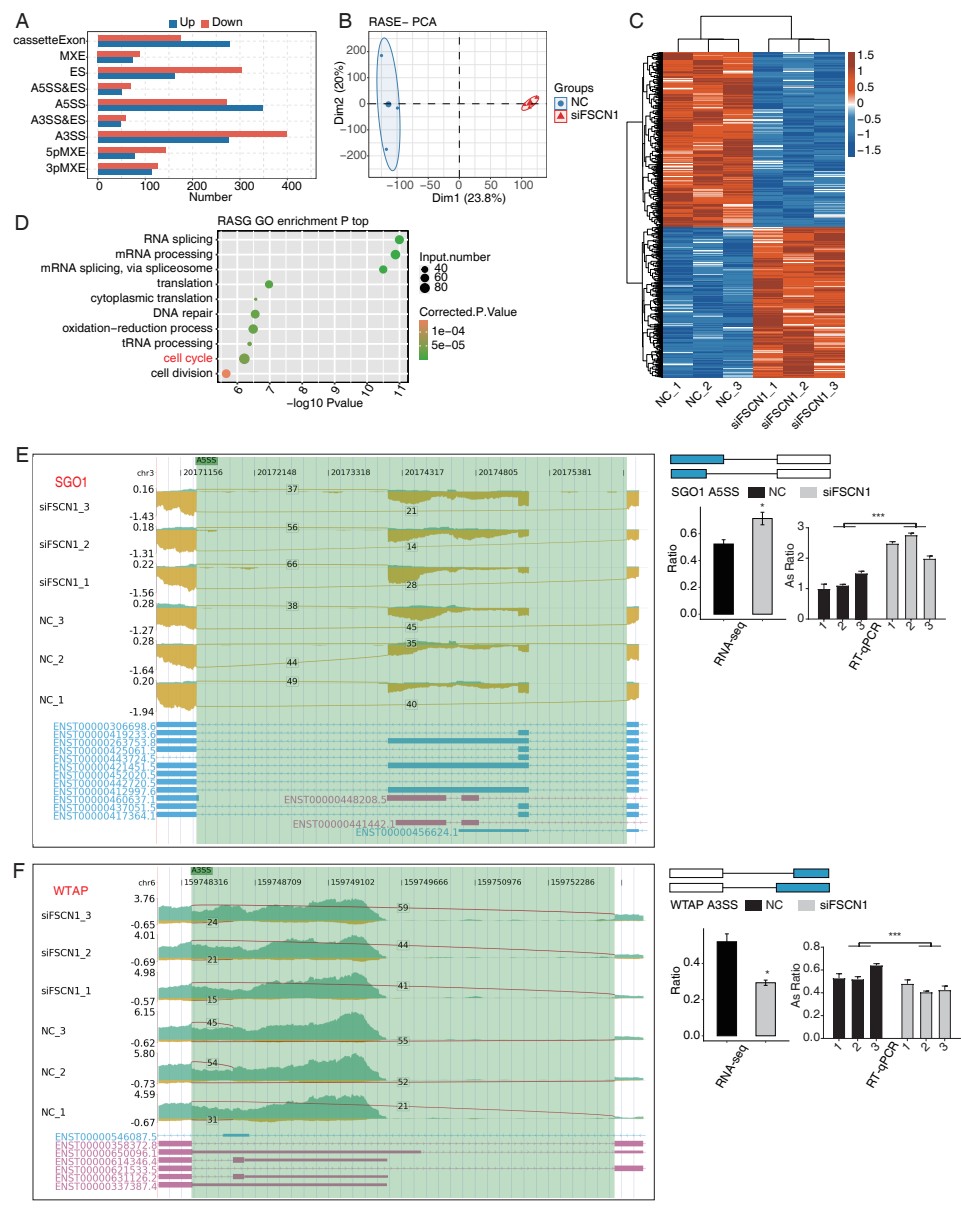

**Figure 4** **FSCN1 regulates the alternative splicing of gene association with cell cycle in A549 cells.** (A) Bar plot showing the FSCN1-regulated alternative splicing event (RASE). (B) PCA based on ratio values of all different expression levels of FSCN1-RASE. (C) Hierarchical clustering heat map of FSCN1-RASE based on ratio values. (D) Scatter plot exhibiting the most enriched GO biological process results of the regulated alternative splicing genes (RASGs). (E) FSCN1 regulates alternative splicing of SGO1. Left panel: IGV-sashimi plot showing the RASEs and binding sites across mRNA. The read distribution of RASE is plotted in the top panel, and the transcripts of each gene are shown below. Right panel: Schematics depicting the structures of ASEs. RNA-seq validation of ASEs is shown in the bottom panel. Reverse transcription qPCR validation of ASEs regulated by FSCN1 in cancer cells. Error bars represent mean ± SEM. (F) FSCN1 regulates alternative splicing of WTAP. Left panel: IGV-sashimi plot showing the RASEs and binding sites across mRNA. The read distribution of RASE is plotted in the top panel, and the transcripts of each gene are shown below. Right panel: Schematic depicting the structures of ASEs. RNA-seq validation *(continued on next page...)*

**Figure 4 (… continued)**
of ASEs are shown in the bottom right panel. Reverse transcription qPCR validation of ASEs regulated by FSCN1 in cancer cells; black bars denote the control group, and grey bars denote the FSCN1-silenced group. Error bars represent mean ± SEM. *$P$-value < 0.05, ***$P$-value < 0.001.

## FSCN1 bound mRNAs and regulation of their expression in A549 cells

We also overlapped the RNA-seq and 242 peak genes. overlap analysis of DEGs and olp peak genes revealed 39 genes that likely bind FSCN1 and undergo differential expression (Fig. 6A). The DEGs-difference ratios and $P$ values observed, combined with the read-distribution map, suggested that FSCN1 may bind to the mRNAs of *ACTG1*, *KRT7*, and *PDE3A* and affect their levels (Figs. 6C–6E). PCR results showed that the expression level of *KRT7* and *ACTG1* genes was decreased in the experimental group, and that of *PDE3A* was increased; two-way ANOVA showed significant differences, as anticipated (Fig. 6B). We also used GEPIA2 and K–M methods to analyze survival times based on TCGA data. K–M analysis revealed that high *ACTG1* and *KRT7* expression predicted poor prognosis in patients with LUAD ($P = 0.0096$ and $P = 0.015$, respectively; Fig. S10). Studies have shown that keratin 7 (KRT7) is a key effector of m$^6$A-induced lung metastasis of breast cancer, which promotes metastasis by increasing the stability of a KRT7-AS/KRT7 mRNA duplex and translation of KRT7 (*Chen et al., 2021*). In addition, exosomal PGAM1 could bind to $\gamma$-actin (ACTG1), which promotes podosome formation and metastasis in pancreatic cancer (*Luo et al., 2023*). This highlights a potential functional mechanism of FSCN1 in NSCLC.

## FSCN1 bound mRNAs and regulated variable splicing in A549 cells

Overlap analysis of genes with differential changes in AS and olp_peak genes revealed 72 genes that likely bind FSCN1 and undergo differential ASEs (Fig. 7A). The ASE-difference ratios and $P$ values observed, combined with the read-distribution map, suggested that FSCN1 may bind *NME4*, *NCOR2*, and *EEF1D* mRNAs and affect their AS levels. PCR results showed that compared with that in the control group, the *EEF1D* gene expression rate decreased and *NCOR2* expression rate increased in the experimental group. Two-way ANOVA showed that the difference was significant, which was consistent with the expected results. The ASE rate of the *NME4* gene in the experimental group was higher than that in the control group, which was contrary to the expectation (Figs. 7B–7C, Figs. S11 and S12). Thus, we conclude that FSCN1 can act as an RNA-binding protein and play an alternative splicing role, which has not been previously reported in NSCLC.

## DISCUSSION

*Liang et al. (2022)* first verified the higher expression of FSCN1 in LUSC than in normal tissues using quantitative reverse transcription PCR (qRT-PCR) and immunohistochemistry. FSCN1 expression in NSCLC tissues positively correlated with differentiation and the TNM classification of NSCLC, and high FSCN1 expression was associated with poor prognosis in patients with NSCLC (*Luo et al., 2015*). In this study, TCGA data analysis revealed high FSCN1 expression in LUSC ($P < 0.05$) but not in LUAD.

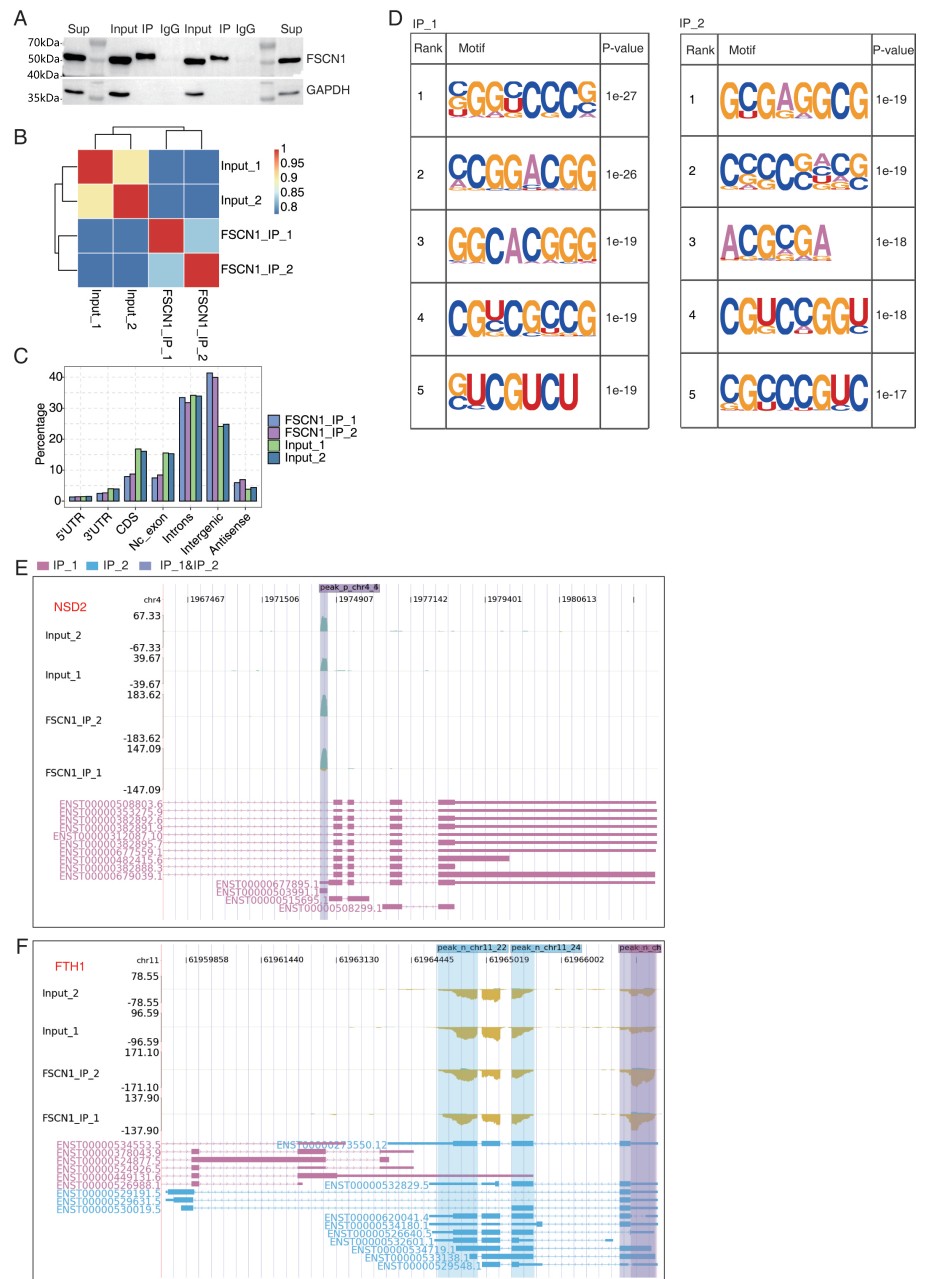

**Figure 5  FSCN1 binds to mRNA associated with lung carcinoma in A549 cells.** (A) Western blot analysis of FSCN1 immunoprecipitates (IPs) using anti-Flag monoclonal antibody. Two replicates were analyzed. (B) Heat map showing the hierarchically clustered Pearson correlation matrix resulting from comparing the transcript expression values for IP and input samples. (C) Bar plot showing the genomic distribution of FSCN1-bound peaks from the two biological replicates. (D) Motif analysis results showing the enriched motifs from FSCN1-bound peaks from the two biological replicates. (E) FSCN1-binding peak genes of *NSD2*. IGV-sashimi plot showing the peaks reads and binding sites across mRNA; the sky blue and reddish purple panels represent the position of peaks. The read distribution of the bound gene is plotted in the top 

**Figure 5 (…continued)**
panel, and the transcripts of each gene are presented in the bottom panel. (F) FSCN1-binding peak genes of *FTH1*. IGV-sashimi plot showing the peak reads and binding sites across mRNA; the sky blue and reddish purple panels represent the position of peaks. The read distribution of the bound gene is plotted and presented in the top panel and the transcripts of each gene are presented in the bottom panel.

High FSCN1 expression was associated with a relatively short overall survival (OS) in patients with LUAD ($P = 0.0016$) but not in patients with LUSC ($P = 0.48$), suggesting that the function of FSCN1 may vary in different lung cancer tissues. Positive FSCN1 expression is detected in 5% of typical carcinoid tumors, 35% of atypical carcinoid tumors, 83% of large-cell neuroendocrine lung cancer, and 100% of small-cell lung cancer (*Pelosi et al., 2003*). Thus, positive FSCN1 expression varies in different pathological tissues in lung tumors, which may indicate an associated tissue-specific regulatory mechanism.

Moreover, the FSCN1 knockdown cell model was constructed using plasmid transfection, and the results suggested that FSCN1 promoted cell apoptosis and suppressed cell proliferation, invasion, and migration abilities of the lung cancer cell. As reported previously, FSCN1 overexpression in A549 cells does not promote their proliferation (*Zhao et al., 2010*). In addition, FSCN1 can promote the proliferation of A549 cells *via* the YAP/TEAD pathway (*Zhao et al., 2018*). Based on these characteristics, we analyzed the potential targets of FSCN1-regulated transcription and alternative splicing levels in A549 cells according to the transcriptome sequencing data. FSCN1 knockdown altered the expression levels and ASEs of numerous genes, suggesting that FSCN1 regulates gene transcription and post-transcriptional events in A549 cells. GO enrichment analysis revealed that downregulated DEGs with increased ASEs were enhanced for cell-cycle-related functions. The mechanism whereby FSCN1 influences tumor migration and invasion is well understood, whereas the mechanism whereby FSCN1 promotes tumor cell proliferation and growth is not. MAPKs are key for cell proliferation-related signaling pathways, and FSCN1 can promote NSCLC by activating the MAPK pathway (*Zhao et al., 2018*). Another study points out that FSCN1 knockdown can inhibit NSCLC cell growth by blocking the YAP/TEAD signaling pathway (*Liang et al., 2016*). We found that FSCN1 can induce numerous cell-cycle-related genes enriched for cell-cycle-signaling pathways, including *CDC25B*, *MCM7*, and *CDK4*. CDC25B is significantly upregulated in squamous cell carcinoma (SCC). CDC25B overexpression can inhibit apoptosis (*Al-Matouq, Holmes & Hansen, 2019*). High expression of MCM7 in NSCLC tissues predicts poor prognosis of patients. Silencing *MCM7* significantly inhibited tumor cell proliferation and could serve as a prognostic marker of NSCLC (*Liu et al., 2017*). In H1299 and H1975 cells, YAP and TAZ can inhibit the cell-cycle regulator p21 by inducing MCM7 transcription, thereby promoting cell proliferation (*Lo Sardo et al., 2017*).

The results of our iRIP-seq experiments showed that FSCN1 may bind ACTG1, KRT7, and PDE3A in A549 cells and affect their mRNA expression levels. Previous research showed that ASAP3 is integral for cell migration, which is crucial for cytoskeleton-network remodeling. In some types of cancer cells, ACTG1 instability can lead to decreased ASAP3 expression, which inhibits cell migration and invasion (*Luo et al., 2014*). PDE3A is

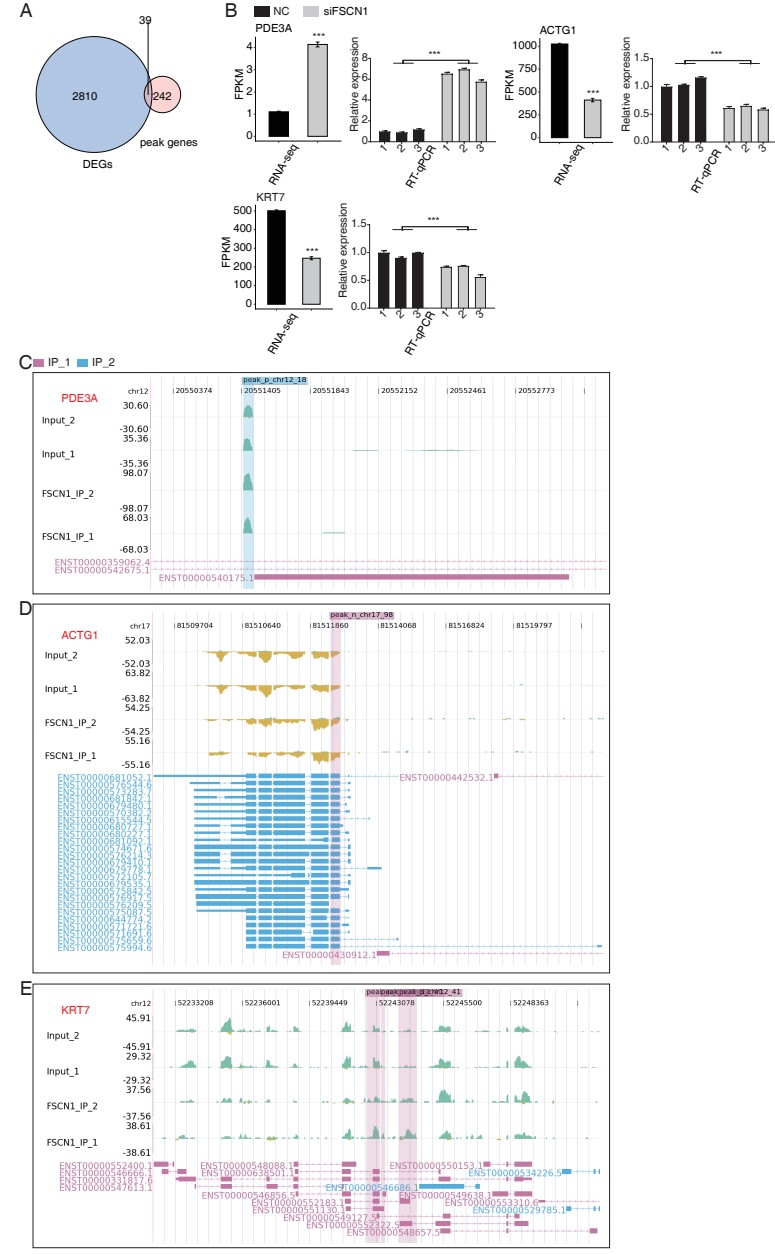

**Figure 6** **FSCN1 binds to RNA and regulates its expression.** (A) Venn diagram showing the overlapped genes between DEGs and peak genes. (B) Bar plot showing the expression pattern and statistical difference of DEGs for the three genes from the overlapped gene set. Reverse transcription qPCR validation of DEGs regulated by FSCN1 in cancer cells; black bars denote the control group, and grey bars denote the FSCN1-silenced group; Error bars represent mean ± SEM. ***P-value < 0.001. (C) FSCN1 binding peak genes of PDE3A. IGV-sashimi plot showing the peaks reads and binding sites across mRNA; the sky blue and red-dish purple panels represent the position of peaks. The read distribution of the bound gene is plotted in the top panel, and (continued on next page...)

**Figure 6 (...continued)**
the transcripts of each gene are shown in the bottom panel. (D) FSCN1-binding peak genes of *ACTG1*. IGV-sashimi plot showing the peaks reads and binding sites across mRNA; the sky blue and reddish purple panels represent the position of peaks. The read distribution of the bound gene is plotted in the top panel, and the transcripts of each gene in the bottom panel. (E) FSCN1-binding peak genes of *KRT7*. IGV-sashimi plot showing the peak reads and binding sites across mRNA; the sky blue and reddish purple panels panels represent the position of peaks. The read distribution of the bound gene is plotted in the top panel, and the transcripts of each gene are shown in the bottom panel.

downregulated in drug-resistant NSCLC cells and enforced PDE3A expression can resist cisplatin in A549 cells. High PDE3A expression improved OS and progression-free survival rates for patients with adenocarcinoma (*Tian et al., 2017*). Relevant studies of these genes suggest that FSCN1 may affect NSCLC progression, regulating their expression. Further *in vitro* studies are required to confirm the molecular mechanism between FSCN1 and cell cycle progression.

We also noted that genes that undergo AS participate in the cell cycle. FSCN1 knockdown in A549 cells may affect the AS of cell-cycle-related genes (*i.e., BCCIP, DLGAP5, PRC1, RECQL5, WTAP*, and *SGO1*). Overexpression of BCCIP is associated with shorter survival time of patients with lung adenocarcinoma, and *in vitro* experiments showed that interference of BCCIP inhibits the proliferation of lung adenocarcinoma and blocks in G1/S phase. Moreover, BCCIP is associated with B cells, macrophages, and dendritic cells (*Shi et al., 2021*). The expression level of DLGAP5 was negatively correlated with the prognosis of patients, and silencing DLGAP5 can cause cell-cycle arrest and inhibit NSCLC cell proliferation (*Love, Huber & Anders, 2014*). PRC1 was expressed at higher levels in NSCLC tissues. PRC1 can promote cell proliferation during the G2/M phase *via* the FAK–paxillin pathway and regulate the phosphorylation of p21/p27-pRB family members (*Liang et al., 2019*). RECQL5 is highly expressed in LUSC and lung adenocarcinoma and can serve as an NSCLC biomarker or clinical target. *In vitro* experiments showed that the expression level of RECQL5 was positively correlated with the occurrence of the malignant phenotype of NSCLC and the expression level of EMT-related proteins (E-cadherin, Snail, MMP2, and MMP9) (*Xia et al., 2021*). High WTAP expression in NSCLC cells was associated with a poor patient prognosis (*Liang et al., 2019*). Previous data showed that the lncRNA PCGEM1 can upregulate WTAP and promote NSCLC progression by adsorbing miR-433-3p (*Cheng et al., 2022*).

Association analysis between mRNA-binding targets identified for FSCN1 and the RNA-seq data generated in this project showed that FSCN1 may bind *NME4, NCOR2,* and *EEF1D* mRNA and affect their pre-mRNA AS levels. Nucleoside diphosphate kinase 4 (NME4) is highly expressed in NSCLC tissues. Knocking down *NME4* in A549 cells resulted in cell cycle arrest cell at the G1 phase, thereby inhibiting cell proliferation and colony formation (*Wang et al., 2019b*). Notably, the PCR verification in this study revealed that the rate of alternative splicing events of *NME4* in the experimental group increased compared with that in the control group, and the specific mechanism warrants further investigation. Low *NCOR2* mRNA expression is positively correlated with shorter OS in LUAD patients, and downregulating *NCOR2* promoted the expression of the tumor factors

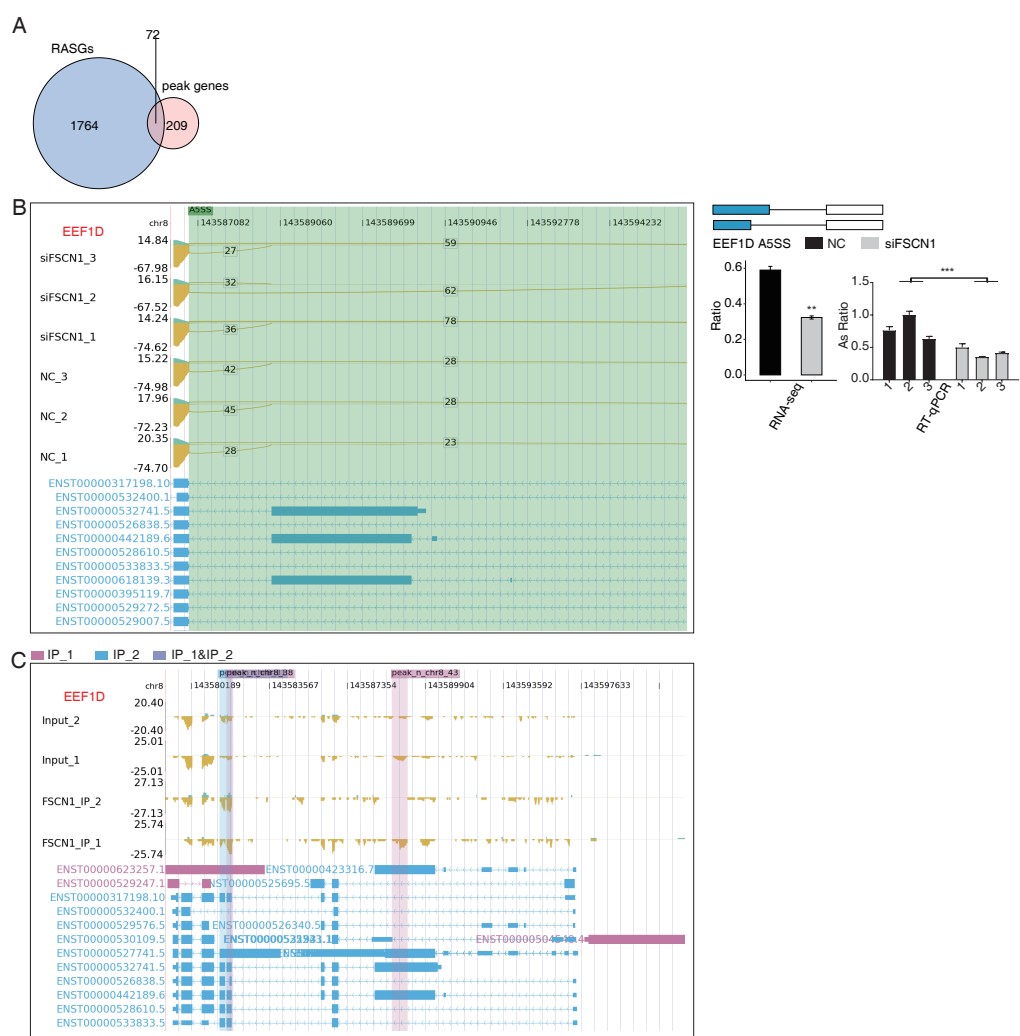

**Figure 7** **FSCN1 binds to mRNA and regulates its alternative splicing.** (A) Venn diagram showing the overlapped genes between RASGs and peak genes. (B) FSCN1 regulates alternative splicing of EEF1D. Left panel: IGV-sashimi plot showing the RASEs and binding sites across mRNA. The read distribution of RASE is plotted in the top panel, and the transcripts of each gene are presented in the bottom panel. Right panel: Schematics depicting the structures of ASEs. RNA-seq validation of ASEs is shown in the bottom right panel. Reverse transcription qPCR validation of ASEs regulated by FSCN1 in cancer cells. Error bars represent mean ± SEM. **P-value < 0.01. (C) FSCN1-binding peak genes of *EEF1D*. IGV-sashimi plot showing the peaks reads and binding sites across mRNA; the sky blue and reddish purple panels represent the position of peaks. The read distribution of the bound gene is plotted in the top panel, and the transcripts of each gene are shown in the bottom panel.

ELK1 and AXL (*Alam et al., 2018*). Here, we found that *EEF1D* was upregulated in human osteosarcoma. *EEF1D* knockdown inhibited the G2/M transition *in vitro* (*Cheng et al., 2018*). These findings suggest that FSCN1 may regulate cell-cycle changes by regulating the AS levels of these target genes in NSCLC.

Previous reports indicated that lncRNAs regulate FSCN1 translation through sponges such as miRNAs. For example, the lncRNA-UCA1 promotes the progression of the

malignant phenotype of bladder cancer *via* the miR-145–FSCN1 pathway (*Xue et al., 2016*). In bladder cancer, ZEB1-AS1 regulated FSCN1 expression through miR-200b sponges (*Gao et al., 2019*). In esophageal cancer, the lncRNA–TTN–AS1 enhanced FSCN1 expression through miR-133b sponges (*Lin et al., 2018*). No reports have shown that lncRNAs directly interact with FSCN1 in NSCLC. We detected abundant levels of FSCN1 bound to lncRNAs (SNHG20 and NEAT1, NSD2, and FTH1). Previous research showed that SNHG20 is highly expressed in NSCLC and can interact with the enhancer of histone methyltransferase 2 to drive P21 expression, thereby inhibiting apoptosis and promoting tumor proliferation and invasion (*Chen et al., 2017*), suggesting that SNHG20 is closely associated with NSCLC. High NEAT expression promoted SULF1 expression by binding to miR-376b-3p, promoting tumor cell growth and apoptosis (*Chen et al., 2020*). NSD2 and FTH1 have not been previously reported in NSCLC.

This study has certain limitations as it did not investigate the function of FSCN1 in other LUAD and LUSC lines. Based on the findings of the present study, the mechanism underlying the role of FSCN1 in NSCLC warrants further investigation.

## CONCLUSIONS

Using RNA-seq technology, we demonstrated that FSCN1 regulates AS, which may regulate tumor proliferation by promoting the expression of cell-cycle-related genes. Our data provide global insights into the regulatory mechanisms mediated by FSCN1 and its target genes in lung cancer.

## ACKNOWLEDGEMENTS

We wish to thank the State Key Laboratory of Central Asia High Incidence and Prevention, Xinjiang Medical University, for providing the experimental platform; our sincere thanks to Ruixue Liu for her help in writing and experiment. We also thank Haiping Zhang, Liang Zong and Xiaoliang Jing for their help with Language, Long Ma and Jie Li for offering help with statistical analyses, and Liwei Zhang for aiding in the final review and proofreading of the manuscript.

### Funding
The study was supported by the State Key Laboratory of Central Asian High Fever Pathogenesis and Prevention, Xinjiang Uygur Autonomous Region (SKL-HIDCA-2020-SG3). The funders had no role in study design, data collection and analysis, decision to publish, or preparation of the manuscript.

### Grant Disclosures
The following grant information was disclosed by the authors:
State Key Laboratory of Central Asian High Fever Pathogenesis and Prevention, Xinjiang Uygur Autonomous Region: SKL-HIDCA-2020-SG3.

## Competing Interests

The authors declare there are no competing interests.

## Author Contributions

- Qingchao Sun conceived and designed the experiments, prepared figures and/or tables, authored or reviewed drafts of the article, and approved the final draft.
- Ruixue Liu conceived and designed the experiments, prepared figures and/or tables, authored or reviewed drafts of the article, and approved the final draft.
- Haiping Zhang performed the experiments, prepared figures and/or tables, and approved the final draft.
- Liang Zong analyzed the data, authored or reviewed drafts of the article, and approved the final draft.
- Xiaoliang Jing analyzed the data, prepared figures and/or tables, and approved the final draft.
- Long Ma performed the experiments, prepared figures and/or tables, and approved the final draft.
- Jie Li performed the experiments, prepared figures and/or tables, and approved the final draft.
- Liwei Zhang conceived and designed the experiments, authored or reviewed drafts of the article, and approved the final draft.

## Data Availability

The data is available at NCBI GEO: GSE234860, GSE234859, GSE234858 and raw measurements are available in the Supplemental File.

## Supplemental Information

Supplemental information for this article can be found online at http://dx.doi.org/10.7717/peerj.16526#supplemental-information.

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
