# Peer review of "Fascin actin-bundling protein 1 regulates non-small cell lung cancer progression by influencing the transcription and splicing of tumorigenesis-related genes"

_PeerJ, doi:10.7717/peerj.16526_

## Round 0.1 · original submission · Minor Revisions

Please address the concerns of both reviewers and amend the manuscript accordingly.

**Language Note:** The review process has identified that the English language must be improved. PeerJ can provide language editing services - please contact us at copyediting@peerj.com for pricing (be sure to provide your manuscript number and title). Alternatively, you should make your own arrangements to improve the language quality and provide details in your response letter. – PeerJ Staff

Reviewer 1 ·

Basic reporting

See additional comments.

Experimental design

See additional comments.

Validity of the findings

See additional comments.

Additional comments

Ref: Manuscript #86706 “Fascin actin-bundling protein 1 regulates non-small cell lung cancer progression by influencing transcription and splicing of tumorigenesis-related genes”



The authors have nicely demonstrated FSCN1 effect on transcription and splicing of genes related to NSCLC disease progression. However, the underlying mechanism/s have not been elucidated, though some possible scenarios were included in the discussion. Some major and minor concerns were included for the authors to address.



Major points:

1. The author have used overall survival Figure 1A. What about progression-free or recurrence free survival? As the authors indicated a difference between FSCN1 and disease progression in LUAD and LUSC.
2. The mechanism that regulate FSCN1 binding and regulation of genes related to disease progression has been discussed but not tested in this study.
3. Figure 1B legend, it was indicated what ** denote and it should be ***
4. * and ** in Figure 1D and 1E, reactively, should be mentioned in the figure legend to be consistent.
5. Figure 2A, the scale bars in RED are not clear.
6. The legend for Figure 2B should indicate what ** and *** denote. Does NC stand for SiRNA? If so, may be it should be replace on the figure, as NC could mean any normal control.
7. Has any of the SiRNA2 and SiRNA3 been used to validate any of the results to rule out any off target effect?
8. Would be very supportive to the author’s conclusions to validate their findings in other lung adenocarcinoma cell line.
9. Could you comment on the predicated results if LUSC cell line was used instead of LUAD?
10. The author should add a summary at the end of each section of the results to explain the results in big picture.



Minor points:

- There are several typos that need to be corrected. See the attached manuscript with track changes turned ON showing the typos that I could spot.
- The manuscript should be carefully read and revised, especially the discussion where some sentences were fragmented.

Annotated reviews are not available for download in order to protect the identity of reviewers who chose to remain anonymous.

Reviewer 2 ·

Basic reporting

In the current study, Sun et al propose a mechanism linking FSCN1 and non-small cell lung
cancer by describing its role as a RNA-binding protein involved in mRNA splicing which affects gene expression including mRNAs involved in regulating cell cycle progression and tumorigenesis. Overall it's a well designed, coherent study presented with sufficient background context and references as well as detailed methodology.
The article includes raw data for each experiment as recommended and the language is overall clear and although there are mistakes and typos so it could be beneficial to include another round of copy editing.

Experimental design

The methodolgy and experimental design appears to be appropriate for the question and the research focus is relevant and within the scope of the journal.

While the methodology and figures are well presented, the data reporting is a bit like an unstructured flood of information, which could potentially be organised better to make a more coherent and interesting dataset. Similarly many of the figure legends (especially earlier figures) are very sparse on information and could be potentially improved.
It's also unclear why in Fig 2C, the cells appear to be different in size between the control and siRNA conditions.

Validity of the findings

The data provided seem to be robust and well integrated with the question and hypothesis set out by the authors.The rationale and benefit for the study is well discussed.

---

## Round 0.2 · accepted · Accept

All comments of the reviewers were adequately addressed and the revised manuscript is acceptable now.

Reviewer 1 ·

Basic reporting

My comments were addressed.

Experimental design

My comments were addressed.

Validity of the findings

My comments were addressed.

Additional comments

No.